# Learning Infinite RBMs with Frank-Wolfe

**Wei Ping**[*]      **Qiang Liu**[†]      **Alexander Ihler**[*]
[*]Computer Science, UC Irvine      [†]Computer Science, Dartmouth College
{wping,ihler}@ics.uci.edu   qliu@cs.dartmouth.edu

## Abstract

In this work, we propose an infinite restricted Boltzmann machine (RBM), whose maximum likelihood estimation (MLE) corresponds to a constrained convex optimization. We consider the Frank-Wolfe algorithm to solve the program, which provides a sparse solution that can be interpreted as inserting a hidden unit at each iteration, so that the optimization process takes the form of a sequence of finite models of increasing complexity. As a side benefit, this can be used to easily and efficiently identify an appropriate number of hidden units during the optimization. The resulting model can also be used as an initialization for typical state-of-the-art RBM training algorithms such as contrastive divergence, leading to models with consistently higher test likelihood than random initialization.

## 1   Introduction

Restricted Boltzmann machines (RBMs) are two-layer latent variable models that use a layer of hidden units $h$ to model the distribution of visible units $v$ [Smolensky, 1986, Hinton, 2002]. RBMs have been widely used to capture complex distributions in numerous application domains, including image modeling [Krizhevsky et al., 2010], human motion capture [Taylor et al., 2006] and collaborative filtering [Salakhutdinov et al., 2007], and are also widely used as building blocks for deep generative models, such as deep belief networks [Hinton et al., 2006] and deep Boltzmann machines [Salakhutdinov and Hinton, 2009]. Due to the intractability of the likelihood function, RBMs are usually learned using the contrastive divergence (CD) algorithm [Hinton, 2002, Tieleman, 2008], which approximates the gradient of the likelihood using a Gibbs sampler.

One practical problem when using a RBM is that we need to decide the size of the hidden layer (number of hidden units) before performing learning, and it can be challenging to decide what is the optimal size. One simple heuristic is to search the 'best' number of hidden units using cross validation or testing likelihood within a pre-defined candidate set. Unfortunately, this is extremely time consuming; it involves running a full training algorithm (e.g., CD) for each possible size, and thus we can only search over a relatively small set of sizes using this approach.

In addition, because the log-likelihood of the RBM is highly non-convex, its performance is sensitive to the initialization of the learning algorithm. Although random initializations (to relatively small values) are routinely used in practice with algorithms like CD, it would be valuable to explore more robust algorithms that are less sensitive to the initialization, as well as smarter initialization strategies to obtain better results.

In this work, we propose a fast, greedy algorithm for training RBMs by inserting one hidden unit at each iteration. Our algorithm provides an efficient way to determine the size of the hidden layer in an adaptive fashion, and can also be used as an initialization for a full CD-like learning algorithm. Our method is based on constructing a convex relaxation of the RBM that is parameterized by a distribution over the weights of the hidden units, for which the training problem can be framed as a convex functional optimization and solved using an efficient Frank-Wolfe algorithm [Frank and Wolfe, 1956, Jaggi, 2013] that effectively adds one hidden unit at each iteration by solving a relatively fast inner loop optimization.

**Related Work**   Our contributions connect to a number of different themes of existing work within machine learning and optimization. Here we give a brief discussion of prior related work.

There have been a number of works on convex relaxations of latent variable models in functional space, which are related to the gradient boosting method [Friedman, 2001]. In supervised learning, Bengio et al. [2005] propose a convex neural network in which the number of hidden units is unbounded and can be learned, and Bach [2014] analyzes the appealing theoretical properties of such a model. For clustering problems, several works on convex functional relaxation have also been proposed [e.g., Nowozin and Bakir, 2008, Bradley and Bagnell, 2009]. Other forms of convex relaxation have also been developed for two layer latent variable models [e.g., Aslan et al., 2013].

There has also been considerable work on extending directed/hierarchical models into "infinite" models such that the dimensionality of the latent space can be automatically inferred during learning. Most of these methods are Bayesian nonparametric models, and a brief overview can be found in Orbanz and Teh [2011]. A few directions have been explored for undirected models, particularly RBMs. Welling et al. [2002] propose a boosting algorithm in the feature space of the model; a new feature is added into the RBM at each boosting iteration, instead of a new hidden unit. Nair and Hinton [2010] conceptually tie the weights of an infinite number of binary hidden units, and connect these sigmoid units with noisy rectified linear units (ReLUs). Recently, Côté and Larochelle [2015] extend an ordered RBM model with infinite number of hidden units, and Nalisnick and Ravi [2015] use the same technique for word embedding. The ordered RBM is sensitive to the ordering of its hidden units and can be viewed as an mixture of RBMs. In contrast, our model incorporates regular RBMs as a special case, and enables model selection for standard RBMs.

The Frank-Wolfe method [Frank and Wolfe, 1956] (a.k.a. conditional gradient) is a classical algorithm to solve constrained convex optimization. It has recently received much attention because it unifies a large variety of sparse greedy methods [Jaggi, 2013], including boosting algorithms [e.g., Beygelzimer et al., 2015], learning with dual structured SVM [Lacoste-Julien et al., 2013] and marginal inference using MAP in graphical models [e.g., Belanger et al., 2013, Krishnan et al., 2015].

Verbeek et al. [2003] proposed a greedy learning algorithm for Gaussian mixture models, which inserts a new component at each step and resembles our algorithm in its procedure. As one benefit, it provides a better initialization for EM than random initialization. Likas et al. [2003] investigate greedy initialization in k-means clustering.

## 2   Background

A restricted Boltzmann machine (RBM) is an undirected graphical model that defines a joint distribution over the vectors of visible units $\boldsymbol{v} \in \{0,1\}^{|\boldsymbol{v}| \times 1}$ and hidden units $\boldsymbol{h} \in \{0,1\}^{|\boldsymbol{h}| \times 1}$,

$$p(\boldsymbol{v}, \boldsymbol{h} \mid \theta) = \frac{1}{Z(\theta)} \exp\left(\boldsymbol{v}^\top W \boldsymbol{h} + \boldsymbol{b}^\top \boldsymbol{v}\right); \quad Z(\theta) = \sum_{\boldsymbol{v}} \sum_{\boldsymbol{h}} \exp\left(\boldsymbol{v}^\top W \boldsymbol{h} + \boldsymbol{b}^\top \boldsymbol{v}\right), \quad (1)$$

where $|\boldsymbol{v}|$ and $|\boldsymbol{h}|$ are the dimensions of $\boldsymbol{v}$ and $\boldsymbol{h}$ respectively, and $\theta := \{W, \boldsymbol{b}\}$ are the model parameters including the pairwise interaction term $W \in \mathbb{R}^{|\boldsymbol{v}| \times |\boldsymbol{h}|}$ and the bias term $\boldsymbol{b} \in \mathbb{R}^{|\boldsymbol{v}| \times 1}$ for the visible units. Here we drop the bias term for the hidden units $\boldsymbol{h}$, since it can be achieved by introducing a dummy visible unit whose value is always one. The partition function $Z(\theta)$ serves to normalize the probability to sum to one, and is typically intractable to calculate exactly.

Because RBMs have a bipartite structure, the conditional distributions $p(\boldsymbol{v}|\boldsymbol{h}; \theta)$ and $p(\boldsymbol{h}|\boldsymbol{v}; \theta)$ are fully factorized and can be calculated in closed form,

$$p(\boldsymbol{h}|\boldsymbol{v}, \theta) = \prod_{i=1}^{|\boldsymbol{h}|} p(h_i|\boldsymbol{v}), \quad \text{with} \quad p(h_i = 1|\boldsymbol{v}) = \sigma\left(\boldsymbol{v}^T W_{\bullet i}\right),$$

$$p(\boldsymbol{v}|\boldsymbol{h}, \theta) = \prod_{j=1}^{|\boldsymbol{v}|} p(v_j|\boldsymbol{h}), \quad \text{with} \quad p(v_j = 1|\boldsymbol{h}) = \sigma\left(W_{j\bullet}\boldsymbol{h} + b_j\right), \quad (2)$$

where $\sigma(u) = 1/(1 + \exp(-u))$ is the logistic function, and $W_{\bullet i}$ and $W_{j\bullet}$ and are the $i$-th column and $j$-th row of $W$ respectively. Eq. (2) allows us to derive an efficient blocked Gibbs sampler that iteratively alternates between drawing $\boldsymbol{v}$ and $\boldsymbol{h}$.

The marginal log-likelihood of the RBM is

$$\log p(\boldsymbol{v} \mid \theta) = \sum_{i=1}^{|\boldsymbol{h}|} \log\left(1 + \exp(\boldsymbol{w}_i^\top \boldsymbol{v})\right) + \boldsymbol{b}^\top \boldsymbol{v} - \log Z(\theta), \tag{3}$$

where $\boldsymbol{w}_i := W_{\bullet i}$ is the $i$-th column of $W$ and corresponds to the weights connected to the $i$-th hidden unit. Because we assume each hidden unit $h_i$ takes values in $\{0, 1\}$, we get the *softplus* function $\log(1 + \exp(\boldsymbol{w}_i^\top \boldsymbol{v}))$ when we marginalize $h_i$. This form shows that the (marginal) free energy of the RBM is a sum of a linear term $\boldsymbol{b}^\top \boldsymbol{v}$ and a set of *softplus* functions with different weights $\boldsymbol{w}_i$; this provides a foundation for our development.

Given a dataset $\{\boldsymbol{v}^n\}_{n=1}^N$, the gradient of the log-likelihood for each data point $\boldsymbol{v}^n$ is

$$\frac{\partial \log p(\boldsymbol{v}^n|\theta)}{\partial W} = \mathbb{E}_{p(\boldsymbol{h}|\boldsymbol{v}^n;\theta)}\left[\boldsymbol{v}^n \boldsymbol{h}^\top\right] - \mathbb{E}_{p(\boldsymbol{v},\boldsymbol{h}|\theta)}\left[\boldsymbol{v}\boldsymbol{h}^\top\right] = \boldsymbol{v}^n(\boldsymbol{\mu}^n)^\top - \mathbb{E}_{p(\boldsymbol{v},\boldsymbol{h}|\theta)}\left[\boldsymbol{v}\boldsymbol{h}^\top\right], \tag{4}$$

where $\boldsymbol{\mu}^n = \sigma(W^\top \boldsymbol{v}^n)$ and the logistic function $\sigma$ is applied in an element-wise manner. The positive part of the gradient can be calculated exactly, since the conditional distribution $p(\boldsymbol{h}|\boldsymbol{v}^n)$ is fully factorized. The negative part arises from the derivatives of the log-partition function and is intractable. Stochastic optimization algorithms, such as CD [Hinton, 2002] and persistent CD [Tieleman, 2008], are popular methods to approximate the intractable expectation using Gibbs sampling.

# 3 RBM with Infinite Hidden Units

In this section, we first generalize the RBM model defined in Eq. (3) to a model with an infinite number of hidden units, which can also be viewed as a convex relaxation of the RBM in functional space. Then, we describe the learning algorithm.

## 3.1 Model Definition

Our general model is motivated by Eq. (3), in which the first term can be treated as an empirical average of the *softplus* function $\log(1 + \exp(\boldsymbol{w}^\top \boldsymbol{v}))$ under an empirical distribution over the weights $\{\boldsymbol{w}_i\}$. To extend this, we define a general distribution $q(\boldsymbol{w})$ over the weight $\boldsymbol{w}$, and replace the empirical averaging with the expectation under $q(\boldsymbol{w})$; this gives the following generalization of an RBM with an infinite (possibly uncountable) number of hidden units,

$$\log p(\boldsymbol{v} \mid q, \vartheta) = \alpha \mathbb{E}_{q(\boldsymbol{w})}\left[\log(1 + \exp(\boldsymbol{w}^\top \boldsymbol{v}))\right] + \boldsymbol{b}^\top \boldsymbol{v} - \log Z(q, \vartheta), \tag{5}$$

$$Z(q, \vartheta) = \sum_{\boldsymbol{v}} \exp\left(\alpha \mathbb{E}_{q(\boldsymbol{w})}\left[\log(1 + \exp(\boldsymbol{w}^\top \boldsymbol{v}))\right] + \boldsymbol{b}^\top \boldsymbol{v}\right),$$

where $\vartheta := \{\boldsymbol{b}, \alpha\}$ and $\alpha > 0$ is a temperature parameter which controls the "effective number" of hidden units in the model, and $\mathbb{E}_{q(\boldsymbol{w})}[f(\boldsymbol{w})] := \int_{\boldsymbol{w}} q(\boldsymbol{w}) f(\boldsymbol{w}) d\boldsymbol{w}$. Note that $q(\boldsymbol{w})$ is assumed to be properly normalized, i.e., $\int_{\boldsymbol{w}} q(\boldsymbol{w}) d\boldsymbol{w} = 1$. Intuitively, (5) defines a semi-parametric model whose log probability is a sum of a linear bias term parameterized by $\boldsymbol{b}$, and a nonlinear term parameterized by the weight distribution $\boldsymbol{w}$ and $\alpha$ that controls the magnitude of the nonlinear term. This model can be regarded as a convex relaxation of the regular RBM, as shown in the following result.

**Proposition 3.1.** *The model in Eq. (5) includes the standard RBM (3) as special case by constraining* $q(\boldsymbol{w}) = \frac{1}{|\boldsymbol{h}|} \sum_{i=1}^{|\boldsymbol{h}|} \mathbb{I}(\boldsymbol{w} = \boldsymbol{w}_i)$ *and* $\alpha = |\boldsymbol{h}|$. *Moreover, the log-likelihood of the model is concave w.r.t. the function* $q(\boldsymbol{w})$, $\alpha$ *and* $\boldsymbol{b}$ *respectively, and is jointly concave with* $q(\boldsymbol{w})$ *and* $\boldsymbol{b}$.

We should point out that the parameter $\alpha$ plays a special role in this model: we reduce to the standard RBM only when $\alpha$ equals the number $|\boldsymbol{h}|$ of particles in $q(\boldsymbol{w}) = \frac{1}{|\boldsymbol{h}|} \sum_{i=1}^{|\boldsymbol{h}|} \mathbb{I}(\boldsymbol{w} = \boldsymbol{w}_i)$, and would otherwise get a *fractional* RBM. The fractional RBM leads to a more challenging inference problem than a standard RBM, since the standard Gibbs sampler is no longer directly applicable. We discuss this point further in Section 3.3.

Given a dataset $\{\boldsymbol{v}^n\}_{n=1}^N$, we learn the parameters $q$ and $\vartheta$ using a penalized maximum likelihood estimator (MLE) that involves a convex functional optimization:

$$\arg\max_{q \in \mathbb{M},\ \vartheta} \left\{ L(q, \vartheta) \equiv \frac{1}{N} \sum_{n=1}^N \log p(\boldsymbol{v}^n \mid q, \vartheta) - \frac{\lambda}{2} \mathbb{E}_{q(\boldsymbol{w})}[||\boldsymbol{w}||^2] \right\}, \tag{6}$$

where $\mathbb{M}$ is the set of valid distributions and we introduce a functional L2 norm regularization $\mathbb{E}_{q(\boldsymbol{w})}[||\boldsymbol{w}||^2]$ to penalize the likelihood for large values of $\boldsymbol{w}$. Alternatively, we could equivalently optimize the likelihood on $\mathbb{M}_C = \{q \mid q(\boldsymbol{w}) \geq 0 \text{ and } \int_{||\boldsymbol{w}||^2 \leq C} q(\boldsymbol{w}) = 1\}$, which restricts the probability mass to a 2-norm ball $||\boldsymbol{w}||^2 \leq C$.

## 3.2 Learning Infinite RBMs with Frank-Wolfe

It is challenging to directly solve the optimization in Eq. (6) by standard gradient descent methods, because it involves optimizing the function $q(\boldsymbol{w})$ with infinite dimensions. Instead, we propose to solve it using the Frank-Wolfe algorithm [Jaggi, 2013], which is projection-free and provides a sparse solution.

Assume we already have $q_t$ at the iteration $t$; then Frank-Wolfe finds the $q_{t+1}$ by maximizing the linearization of the objective function :

$$q_{t+1} \leftarrow (1 - \beta_{t+1})q_t + \beta_{t+1}r_{t+1}, \quad \text{where} \quad r_{t+1} \leftarrow \arg\max_{q \in \mathbb{M}} \langle q, \nabla_q L(q_t, \vartheta_t) \rangle, \qquad (7)$$

where $\beta_{t+1} \in [0, 1]$ is a step size parameter, and the convex combination step guarantees the new $q_{t+1}$ remains a distribution after the update. A typical step-size is $\beta_t = 1/t$, in which case we have $q_t(\boldsymbol{w}) = \frac{1}{t}\sum_{s=1}^{t} r_s(\boldsymbol{w})$, that is, $q_t$ equals the average of all the earlier solutions obtained by the linear program.

To apply Frank-Wolfe to solve our problem, we need to calculate the functional gradient $\nabla_q L(q_t, \vartheta_t)$ in E.q. (7). We can show that (see details in Appendix),

$$\nabla_q L(q_t, \vartheta_t) = -\frac{\lambda}{2}||\boldsymbol{w}||^2 + \alpha_t \left[ \frac{1}{N} \sum_{n=1}^{N} \log(1 + \exp(\boldsymbol{w}^\top \boldsymbol{v}^n)) - \sum_{\boldsymbol{v}} p(\boldsymbol{v} \mid q_t, \vartheta_t) \log(1 + \exp(\boldsymbol{w}^\top \boldsymbol{v})) \right],$$

where $p(\boldsymbol{v} \mid q_t, \vartheta_t)$ is the distribution parametrized by the weight density $q_t(\boldsymbol{w})$ and parameter $\vartheta_t$,

$$p(\boldsymbol{v} \mid q_t, \vartheta_t) = \frac{\exp\left(\alpha_t \mathbb{E}_{q_t(\boldsymbol{w})}[\log(1 + \exp(\boldsymbol{w}^\top \boldsymbol{v}))] + \boldsymbol{b}_t^\top \boldsymbol{v}\right)}{Z(q_t, \vartheta_t)}. \qquad (8)$$

The (functional) linear program in Eq. (7) is equivalent to an optimization over weight vector $\boldsymbol{w}$ :

$$\max_{q \in \mathbb{M}} \langle q, \nabla_q L(q_t, \vartheta_t) \rangle = \max_{q \in \mathbb{M}} \mathbb{E}_{q(\boldsymbol{w})}[\nabla_q L(q_t, \vartheta_t)]$$

$$= -\min_{\boldsymbol{w}} \left\{ \frac{\lambda}{2}||\boldsymbol{w}||^2 + \sum_{\boldsymbol{v}} p(\boldsymbol{v} \mid q_t, \vartheta_t) \log(1 + \exp(\boldsymbol{w}^\top \boldsymbol{v})) - \frac{1}{N} \sum_{n=1}^{N} \log(1 + \exp(\boldsymbol{w}^\top \boldsymbol{v}^n)) \right\} \quad (9)$$

The gradient of the objective (9) is,

$$\nabla_{\boldsymbol{w}} = \lambda \boldsymbol{w} + \mathbb{E}_{p(\boldsymbol{v}|q_t, \vartheta_t)}[\sigma(\boldsymbol{w}^\top \boldsymbol{v}) \cdot \boldsymbol{v}] - \frac{1}{N} \sum_{n=1}^{N} \sigma(\boldsymbol{w}^\top \boldsymbol{v}^n) \cdot \boldsymbol{v}^n,$$

where the expectation over $p(\boldsymbol{v} \mid q_t, \vartheta_t)$ can be intractable to calculate, and one may use stochastic optimization and draw samples using MCMC. Note that the second two terms in the gradient enforce an intuitive moment matching condition: the optimal $\boldsymbol{w}$ introduces a set of "importance weights" $\sigma(\boldsymbol{w}^\top \boldsymbol{v})$ that adjust the empirical data and the previous model, such that their moments match with each other.

Now, suppose $\boldsymbol{w}_t^*$ is the optimum of Eq. (9) at iteration $t$, the item $r_t(\boldsymbol{w})$ we added can be shown to be the delta over $\boldsymbol{w}_t^*$, that is, $r_t(\boldsymbol{w}) = \mathbb{I}(\boldsymbol{w} = \boldsymbol{w}_t^*)$; in addition, we have $q_t(\boldsymbol{w}) = \frac{1}{t}\sum_{s=1}^{t} \mathbb{I}(\boldsymbol{w} = \boldsymbol{w}_s^*)$ when the step size is taken to be $\beta_t = \frac{1}{t}$. Therefore, this Frank-Wolfe update can be naturally interpreted as greedily inserting a hidden unit into the current model $p(\boldsymbol{v} \mid q_t, \vartheta_t)$. In particular, if we update the temperature parameter as $\alpha_t \leftarrow t$, according to Proposition 3.1, we can directly transform our model $p(\boldsymbol{v} \mid q_t, \vartheta_t)$ to a regular RBM after each Frank-Wolfe step, which enables the convenient blocked Gibbs sampling for inference.

Compared with the (regularized) MLE of the standard RBM (e.g. in Eq. (4)), the optimization in Eq. (9) has the following nice properties: (1) The current model $p(\boldsymbol{v} \mid q_t, \vartheta_t)$ does not depend on

---

**Algorithm 1** Frank-Wolfe Learning Algorithm

---

**Input:** training data $\{\boldsymbol{v}^n\}_{n=1}^N$; step size $\eta$; regularization $\lambda$.
**Output:** sparse solution $q^*(\boldsymbol{w})$, and $\vartheta^*$

Initialize $q_0(\boldsymbol{w}) = \mathbb{I}(\boldsymbol{w} = \boldsymbol{w}')$ at random $\boldsymbol{w}'$; $\boldsymbol{b}_0 = 0$; $\alpha_0 = 1$;

**for** $t = 1 : T$ [or, stopping criterion] **do**

    Draw sample $\{\boldsymbol{v}^s\}_{s=1}^S$ from $p(\boldsymbol{v} \mid q_{t-1}, \vartheta_{t-1})$;

    $\boldsymbol{w}_t^* = \operatorname{argmin}_{\boldsymbol{w}} \left\{ \frac{\lambda}{2}||\boldsymbol{w}||^2 + \frac{1}{S}\sum_{s=1}^S \log(1+\exp(\boldsymbol{w}^\top \boldsymbol{v}^s)) - \frac{1}{N}\sum_{n=1}^N \log(1+\exp(\boldsymbol{w}^\top \boldsymbol{v}^n)) \right\}$;

    Update $q_t(\boldsymbol{w}) \leftarrow (1-\frac{1}{t}) \cdot q_{t-1}(\boldsymbol{w}) + \frac{1}{t} \cdot \mathbb{I}(\boldsymbol{w} = \boldsymbol{w}_t^*)$;

    Update $\alpha_t \leftarrow t$ (optional: gradient descent);

    Set $\boldsymbol{b}_t = \boldsymbol{b}_{t-1}$;
    **repeat**
        Draw a mini-batch samples $\{\boldsymbol{v}^m\}_{m=1}^M$ from $p(\boldsymbol{v} \mid q_t, \vartheta_t)$
        Update $\boldsymbol{b}_t \leftarrow \boldsymbol{b}_t + \eta \cdot (\frac{1}{N}\sum_{n=1}^N \boldsymbol{v}^n - \frac{1}{M}\sum_{m=1}^M \boldsymbol{v}^m)$
    **until**

**end for**

Return $q^*(\boldsymbol{w}) = q_t(\boldsymbol{w})$; $\vartheta^* = \{\boldsymbol{b}_t, \alpha_t\}$;

---

$\boldsymbol{w}$, which means we can draw enough samples from $p(\boldsymbol{v} \mid q_t, \vartheta_t)$ at each iteration $t$, and reuse them during the optimization of $\boldsymbol{w}$. (2) The objective function in Eq. (9) can be evaluated explicitly given a set of samples, and hence efficient off-the-shelf optimization tools such as L-BFGS can be used to solve the optimization very efficiently. (3) Each iteration of our method involves much fewer parameters (only the weights for a single hidden unit, which is $|\boldsymbol{v}| \times 1$ instead of the full $|\boldsymbol{v}| \times |\boldsymbol{h}|$ weight matrix are updated), and hence defines a series of easier problems that can be less sensitive to initialization. We note that a similar greedy learning strategy has been successfully applied for learning mixture models [Verbeek et al., 2003], in which one greedily inserts a component at each step, and that this approach can provide better initialization for EM optimization than using multiple random initializations.

Once we obtain $q_{t+1}$, we can update the bias parameter $\boldsymbol{b}_t$ by gradient descent,

$$\nabla_{\boldsymbol{b}} L(q_{t+1}, \vartheta_t) = \frac{1}{N}\sum_{n=1}^N \boldsymbol{v}^n - \sum_{\boldsymbol{v}} p(\boldsymbol{v}|q_{t+1}, \vartheta_t)\boldsymbol{v}. \tag{10}$$

One can further optimize $\alpha_t$ by gradient descent,[1] but we find simply updating $\alpha_t \leftarrow t$ is more efficient and works well in practice. We summarize our Frank-Wolfe learning algorithm in Algorithm 1.

*Adding hidden units on RBM.* Besides initializing $q(\boldsymbol{w})$ to be a delta function at some random $\boldsymbol{w}'$ and learning the model from scratch, one can also adapt Algorithm 1 to incrementally add hidden units into an existing RBM in Eq. (3) (e.g. have been learned by CD). According to Proposition 3.1, one can simply initialize $q_t(\boldsymbol{w}) = \frac{1}{|\boldsymbol{h}|}\sum_{i=1}^{|\boldsymbol{h}|} \mathbb{I}(\boldsymbol{w} = \boldsymbol{w}_i)$, $\alpha_t = |\boldsymbol{h}|$, and continue the Frank-Wolfe iterations at $t = |\boldsymbol{h}| + 1$.

*Removing hidden units.* Since the hidden units are added in a greedy manner, one may want to remove an old hidden unit during the Frank-Wolfe learning, provided it is bad with respect to our objective Eq. (9) after more hidden units have been added. A variant of Frank-Wolfe with *away-steps* [Guélat and Marcotte, 1986] fits this requirement and can be directly applied. As shown by [Clarkson, 2010], it can improve the sparsity of the final solution (i.e., fewer hidden units in the learned model).

### 3.3 MCMC Inference for Fractional RBMs

As we point out in Section 3.1, we need to take $\alpha$ equal to the number of particles in $q(\boldsymbol{w})$ (that is, $\alpha_t \leftarrow t$ in Algorithm 1) in order to have our model reduce to the standard RBM. If $\alpha$ takes a more general real number, we obtain a more general *fractional* RBM model, for which inference is

more challenging because the standard block Gibbs sampler is not directly applicable. In practice, we find that setting $\alpha_t \leftarrow t$ to correspond to a regular RBM seems to give the best performance, but for completeness, we discuss the fractional RBM in more detail in this section, and propose a Metropolis-Hastings algorithm to draw samples from the fractional RBM. We believe that this fractional RBM framework provides an avenue for further improvements in future work.

To frame the problem, let us assume $\alpha q(\boldsymbol{w}) = \sum_i c_i \cdot \mathbb{I}(\boldsymbol{w} = \boldsymbol{w}_i)$, where $c_i$ is a general real number; the corresponding model is

$$\log p(\boldsymbol{v} \mid q, \vartheta) = \sum_i c_i \log(1 + \exp(\boldsymbol{w}_i^\top \boldsymbol{v})) + \boldsymbol{b}^\top \boldsymbol{v} - \log Z(q, \vartheta), \tag{11}$$

which differs from the standard RBM in (3) because each *softplus* function is multiplied by $c_i$. Nevertheless, one may push the $c_i$ into the softplus function, and obtain a standard RBM that forms an approximation of (11):

$$\log \widetilde{p}(\boldsymbol{v} \mid q, \vartheta) = \sum_i \log(1 + \exp(c_i \cdot \boldsymbol{w}_i^\top \boldsymbol{v})) + \boldsymbol{b}^\top \boldsymbol{v} - \log \widetilde{Z}(q, \vartheta). \tag{12}$$

This approximation can be justified by considering the special case when the magnitude of the weights $\boldsymbol{w}$ is very large, so that the softplus function essentially reduces to a ReLU function, that is, $\log(1 + \exp(\boldsymbol{w}_i^\top \boldsymbol{v})) \approx \max(0, \boldsymbol{w}_i^\top \boldsymbol{v})$. In this case, (11) and (12) become equivalent because $c_i \max(0, x) = \max(0, c_i x)$. More concretely, we can guarantee the following bound:

**Proposition 3.2.** *For any $0 < c_i \leq 1$, we have*

$$\frac{1}{2^{1-c_i}}(1 + \exp(c_i \cdot \boldsymbol{w}_i^\top \boldsymbol{v})) \leq (1 + \exp(\boldsymbol{w}_i^\top \boldsymbol{v}))^{c_i} \leq 1 + \exp(c_i \cdot \boldsymbol{w}_i^\top \boldsymbol{v}).$$

The proof can be found in the Appendix. Note that we apply the bound when $c_i > 1$ by splitting $c_i$ into the sum of its integer part and fractional remainder, and apply the bound to the fractional part.

Therefore, the fractional RBM (11) can be well approximated by the standard RBM (12), and this can be leveraged to design an inference algorithm for (11). As one example, we can use the Gibbs update of (12) as a proposal for a Metropolis-Hastings update for (11). To be specific, given a configuration $\boldsymbol{v}$, we perform Gibbs update in RBM $\widetilde{p}(\boldsymbol{v} \mid q, \vartheta)$ to get $\boldsymbol{v}'$, and accept it with probability $\min(1, A(\boldsymbol{v} \to \boldsymbol{v}'))$,

$$A(\boldsymbol{v} \to \boldsymbol{v}') = \frac{p(\boldsymbol{v}')\widetilde{T}(\boldsymbol{v}' \to \boldsymbol{v})}{p(\boldsymbol{v})\widetilde{T}(\boldsymbol{v} \to \boldsymbol{v}')},$$

where $\widetilde{T}(\boldsymbol{v} \to \boldsymbol{v}')$ is the Gibbs transition of RBM $\widetilde{p}(\boldsymbol{v} \mid q, \vartheta)$. Because the acceptance probability of a Gibbs sampler equals one, we have $\frac{\widetilde{p}(\boldsymbol{v})\widetilde{T}(\boldsymbol{v} \to \boldsymbol{v}')}{\widetilde{p}(\boldsymbol{v}')\widetilde{T}(\boldsymbol{v}' \to \boldsymbol{v})} = 1$. This gives

$$A(\boldsymbol{v} \to \boldsymbol{v}') = \frac{p(\boldsymbol{v}')\widetilde{p}(\boldsymbol{v})}{p(\boldsymbol{v})\widetilde{p}(\boldsymbol{v}')} = \frac{\prod_i(1 + \exp(\boldsymbol{w}_i^\top \boldsymbol{v}'))^{c_i} \cdot \prod_i(1 + \exp(c_i \cdot \boldsymbol{w}_i^\top \boldsymbol{v}))}{\prod_i(1 + \exp(\boldsymbol{w}_i^\top \boldsymbol{v}))^{c_i} \cdot \prod_i(1 + \exp(c_i \cdot \boldsymbol{w}_i^\top \boldsymbol{v}'))}.$$

## 4 Experiments

In this section, we test the performance of our Frank-Wolfe (FW) learning algorithm on two datasets: MNIST [LeCun et al., 1998] and Caltech101 Silhouettes [Marlin et al., 2010]. The MNIST handwritten digits database contains 60,000 images in the training set and 10,000 test set images, where each image $\boldsymbol{v}^n$ includes $28 \times 28$ pixels and is associated with a digit label $y^n$. We binarize the grayscale images by thresholding the pixels at 127, and randomly select 10,000 images from training as the validation set. The Caltech101 Silhouettes dataset [Marlin et al., 2010] has 8,671 images with $28 \times 28$ binary pixels, where each image represents objects silhouette and has a class label (overall 101 classes). The dataset is divided into three subsets: 4,100 examples for training, 2,264 for validation and 2,307 for testing.

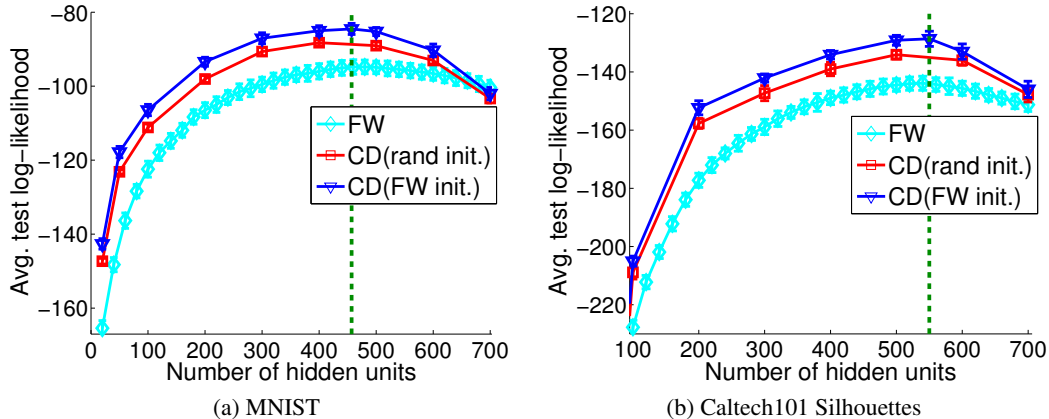

Figure 1: Average test log-likelihood on the two datasets as we increase the number of hidden units. We can see that FW can correctly identify an appropriate hidden layer size with high test log-likelihood (marked by the green dashed line). In addition, CD initialized by FW gives higher test likelihood than random initialization for the same number of hidden units. Best viewed in color.

**Training algorithms**   We train RBMs with CD-10 algorithm. [2] A fixed learning rate is selected from the set $\{0.05, 0.02, 0.01, 0.005\}$ using the validation set, and the mini-batch size is selected from the set $\{10, 20, 50, 100, 200\}$. We use 200 epochs for training on MINIST and 400 epochs on Caltech101. Early stopping is applied by monitoring the difference of average log-likelihood between training and validation data, so that the intractable log-partition function is cancelled [Hinton, 2010]. We train RBMs with $\{20, 50, 100, 200, 300, 400, 500, 600, 700\}$ hidden units. We incrementally train a RBM model by Frank-Wolfe (FW) algorithm 1. A fixed step size $\eta$ is selected from the set $\{0.05, 0.02, 0.01, 0.005\}$ using the validation data, and a regularization strength $\lambda$ is selected from the set $\{1, 0.5, 0.1, 0.05, 0.01\}$. We set $T = 700$ in Algorithm 1, and use the same early stopping criterion as CD. We randomly initialize the CD algorithm 5 times and select the best one on the validation set; meanwhile, we also initialize CD by the model learned from Frank-Wolfe.

**Test likelihood**   To evaluate the test likelihood of the learned models, we estimate the partition function using annealed importance sampling (AIS) [Salakhutdinov and Murray, 2008]. The temperature parameter is selected following the standard guidance: first 500 temperatures spaced uniformly from 0 to 0.5, and 4,000 spaced uniformly from 0.5 to 0.9, and 10,000 spaced uniformly from 0.9 to 1.0; this gives a total of 14,500 intermediate distributions. We summarize the averaged test log-likelihood of MNIST and Caltech101 Silhouettes in Figure 1, where we report the result averaged over 500 AIS runs in all experiments, with the error bars indicating the 3 standard deviations of the estimations.

We evaluate the test likelihood of the model in FW after adding every 20 hidden units. We perform early stopping when the gap of average log-likelihood between training and validation data largely increases. As shown in Figure 1, this procedure selects 460 hidden units on MNIST (as indicated by the green dashed lines), and 550 hidden units on Caltech101; purely for illustration purposes, we continue FW in the experiment until reaching $T = 700$ hidden units. We can see that the identified number of hidden units roughly corresponds to the maximum of the test log-likelihood of all the three algorithms, suggesting that FW can identify the appropriate number of hidden units during the optimization.

We also use the model learned by FW as an initialization for CD (the blue lines in Figure 2), and find it consistently performs better than the best result of CD with 5 random initializations. In our implementation, the running time of the FW procedure is at most twice as CD for the same number of hidden units. Therefore, FW initialized CD provides a practical strategy for learning RBMs: it requires approximately three times of computation time as a single run of CD, while simultaneously identifying the proper number of hidden units and obtaining better test likelihood.

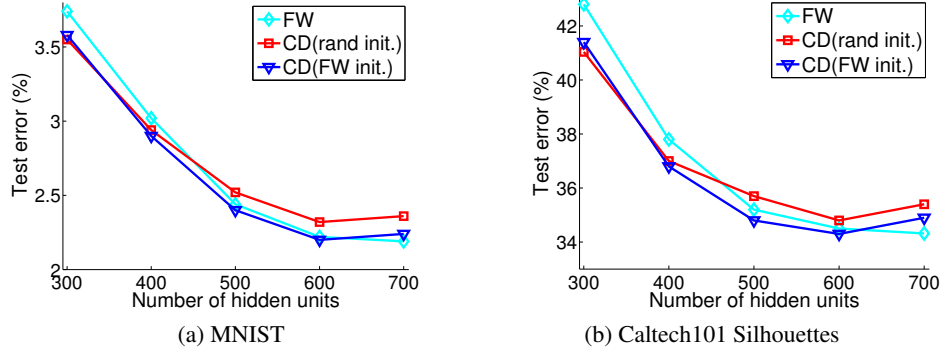

Figure 2: Classification error when using the learned hidden representations as features.

**Classification**    The performance of our method is further evaluated using discriminant image classification tasks. We take the hidden units' activation vectors $\mathbb{E}_{p(\boldsymbol{h}|\boldsymbol{v}^n)}[\boldsymbol{h}]$ generated by the three algorithms in Figure 1 and use it as the feature in a multi-class logistic regression on the class labels $y^n$ in MNIST and Caltech101. From Figure 2, we find that our basic FW tends to be worse than the fully trained CD (best in 5 random initializations) when only small numbers of hidden units are added, but outperforms CD when more hidden units (about 450 in both cases) are added. Meanwhile, the CD initialized by FW outperforms CD using the best of 5 random initializations.

## 5    Conclusion

In this work, we propose a convex relaxation of the restricted Boltzmann machine with an infinite number of hidden units, whose MLE corresponds to a constrained convex program in a function space. We solve the program using Frank-Wolfe, which provides a sparse greedy solution that can be interpreted as inserting a single hidden unit at each iteration. Our new method allows us to easily identify the appropriate number of hidden units during the progress of learning, and can provide an advanced initialization strategy for other state-of-the-art training methods such as CD to achieve higher test likelihood than random initialization.

### Acknowledgements

This work is sponsored in part by NSF grants IIS-1254071 and CCF-1331915. It is also funded in part by the United States Air Force under Contract No. FA8750-14-C-0011 under the DARPA PPAML program.

## Appendix

**Derivation of gradients**    The functional gradient of $L(q, \vartheta)$ w.r.t. the density function $q(\boldsymbol{w})$ is

$$\nabla_q L(q, \vartheta) = -\frac{\lambda}{2}||\boldsymbol{w}||^2 + \alpha \Bigg[ \frac{1}{N} \sum_{n=1}^{N} \log(1 + \exp(\boldsymbol{w}^\top \boldsymbol{v}^n))$$

$$- \frac{\sum_{\boldsymbol{v}} \exp\left(\alpha \mathbb{E}_{q(\boldsymbol{w})}[\log(1 + \exp(\boldsymbol{w}^\top \boldsymbol{v}))] + \boldsymbol{b}^\top \boldsymbol{v}\right) \cdot \log(1 + \exp(\boldsymbol{w}^\top \boldsymbol{v}))}{Z(q, \boldsymbol{b}, \alpha)} \Bigg]$$

$$= -\frac{\lambda}{2}||\boldsymbol{w}||^2 + \alpha \Bigg[ \frac{1}{N} \sum_{n=1}^{N} \log(1 + \exp(\boldsymbol{w}^\top \boldsymbol{v}^n)) - \sum_{\boldsymbol{v}} p(\boldsymbol{v} \mid q, \vartheta) \log(1 + \exp(\boldsymbol{w}^\top \boldsymbol{v})) \Bigg].$$

The gradient of $L(q, \vartheta)$ w.r.t. the temperature parameter $\alpha$ is

$$\nabla_\alpha L(q, \vartheta) = \frac{1}{N} \sum_{n=1}^{N} \mathbb{E}_{q(\boldsymbol{w})}\big[\log(1 + \exp(\boldsymbol{w}^\top \boldsymbol{v}^n))\big] - \sum_{\boldsymbol{v}} p(\boldsymbol{v} \mid q, \vartheta) \, \mathbb{E}_{q(\boldsymbol{w})}\big[\log(1 + \exp(\boldsymbol{w}^\top \boldsymbol{v}))\big].$$

**Proof of Proposition 4.2**

*Proof.* For any $0 < c \leq 1$, we have following classical inequality,

$$\sum_k x_k \leq (\sum_k x_k^c)^{1/c}, \text{ and } \frac{1}{2}\sum_k x_k \leq (\frac{1}{2}\sum_k x_k^c)^{1/c}$$

Let $x_1 = 1$ and $x_2 = \exp(\boldsymbol{w}_i^\top \boldsymbol{v})$, and the proposition is a direct result of above two inequalities. $\square$

## Footnotes

[1]see Appendix for the definition of $\nabla_\alpha L(q_t, \vartheta_t)$

[2] CD-k refers to using k-step Gibbs sampler to approximate the gradient of the log-partition function.

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
