[Reviews · NeurIPS 2016]

Reviewer 1

Summary

The paper proposes an algorithm for unsupervised learning of binary valued MRFs on bipartite graphs (Restricted Boltzmann Machines). The main idea is to represent the probability distribution for the variables in visible layer via a distribution on the parameter space of hidden units, thus allowing for models with an infinite number of hidden units. The resulting learning task can be solved by a Frank-Wolfe algorithm. Each iteration thereof can be interpreted as greedily adding a new hidden unit to the model. Compared to standard methods, this approach has several advantages, although it also requires sampling for estimating the gradient of the objective function in each iteration of the Frank-Wolfe algorithm.

Qualitative Assessment

The paper is well written and technically correct. Experiments show that the proposed method is competitive to standard learning methods (e.g. contrastive divergence) and provides an estimate of the necessary number of hidden units at the same time. Conceptually, the proposed learning approach can be seen as as a variant of boosting, however, in the context of unsupervised learning. This is to my knowledge novel and promising.

Confidence in this Review

2-Confident (read it all; understood it all reasonably well)


Reviewer 2

Summary

This paper describes a greedy method for adding hidden nodes to a restricted Boltzmann machine (RBM). The method is derived by taking the marginal likelihood formula for the RBM, which involves a discrete sum of softplus functions using the weight vectors of the hidden units, and relaxing it to take an expectation with respect to an arbitrary distribution over weight vectors. Optimizing this distribution using a method similar to functional gradient boosting yields the greedy algorithm for adding nodes. The experimental results show that the resulting RBM benefits from further tuning using a more standard learning algorithm such as contrastive divergence, but the proposed algorithm is useful for finding the optimal number of hidden units, and provides a better-than-random initialization point for the RBM weights.

Qualitative Assessment

This paper is well-written and addresses the important practical issue of choosing the number of hidden nodes in an RBM. The application of the Frank-Wolfe algorithm is not particularly novel, given the large set of related work that the paper cites. But the paper makes the key technical move of replacing the discrete sum over hidden nodes in the marginal likelihood function with an arbitrary distribution over weight vectors. This opens the path to applying Frank-Wolfe in a functional gradient setting. I found this move interesting and novel (although there may be related work I don't know about); it might suggest similar approaches for other undirected models besides RBMs. My main question is whether there are other techniques or heuristics that could be included for comparison in the experiments, besides the basic technique of training with various numbers of hidden nodes and various random initializations. For instance, would it be feasible to compare to the ordered RBM approach of Cote and Larochelle? Minor comments: * p. 1, line 37: "a efficient" should be "an efficient". * p. 4, line 140: "the second two terms in the gradient enforces". Should be "enforce". * p. 5, line 169: "one can simply initialize...and continues". Should be "continue". * p. 7, line 209: "CD-10". Would be helpful to say what the 10 means here. * p. 7, line 218: "We randomly initialize the CD algorithm 5 times and select the best one". The best one on the validation set, not the test set, right? Would be good to make this explicit. * p. 7, line 227: "We evaluate the test likelihood of the model in FW after adding every 20 hidden units, and perform early stopping...". The rest of this sentence suggests that the early stopping criterion is based just on the training and validation sets, not the test set, but the beginning of the sentence suggests that the test set is somehow involved. It would be clearer to make these two separate sentences. * p. 8, Figure 2: I'm slightly color-blind and find it very hard to distinguish between the magenta line and the red line in these graphs. Maybe use cyan instead of magenta, and use different marker shapes for the three series. * p. 8, line 254: "to higher test likelihood". A verb is missing here; maybe "to achieve higher test likelihood". == Update After Author Feedback == I appreciate the authors' responses to my comments. It's interesting to hear that the iRBM tends to learn many more hidden units than the proposed method. How does the accuracy compare between the two approaches? I hope it will be feasible to include some comparison in the final version.

Confidence in this Review

2-Confident (read it all; understood it all reasonably well)


Reviewer 3

Summary

This paper proposes a novel approach to learning restricted Boltzman Machines (RBMs) with an unbounded number of hidden units. The main idea is to use a distribution over weight vectors, instead of having explicit hidden units. In practice this distribution is a set of point masses over a finite number of weight vectors. A Frank-Wolfe (FW) algorithm is used to maximize a penalized log-likelihood criterion, which adds one point mass in ever iteration. For a particular setting of a parameter alpha, the resulting model is equivalent to an RBM in each iteration. Experimental results on a generative modeling (density estimation) task on two datasets are provided to show that (1) the approach is about as fast as constrastive divergence (CD) training (standard for RBMs), (2) that CD on top of the FW solution gives better results than CD from random initialization, and (3) that the sequence of FW solutions can be effectively used to select a number of hidden units that is close to optimal.

Qualitative Assessment

This paper presents an interesting and novel approach to learning RBMs with an unbounded number of hidden units. A greedy incremental learning algorithm is developed by means of FW optimization. This is a technically elegant approach, in particular in the case when parameter alpha is chosen to be equal to the iteration number. I believe this paper makes a solid nips paper, which I would recommend for poster presentation given the limited scope of the impact of the contribution of this paper. - One thing that was not clear to me is the impact of re-setting alpha each iteration in this manner has on the log-likelihood. Can setting alpha in this manner negatively impact the log-likelihood? Can we measure this in such a case? This is the only part of the presented approach that seems heuristic. It would be good to be a little clearer on this point. The paper is well written and well organized, generally easy to follow except for some of the technical material. Presentation of the experimental data is clear with sufficient level of detail. References are complete and well formatted. The impact of this paper is probably significant for those interested in RBMs, but this is a limited community. Ideas from this paper might carry over to more general settings, but it is not immediately obvious that this will be the case. The novelty of this work is related to the technical development of the FW algorithm, and also to the conceptual level idea of using a distribution over weights to formulate the case of an unbounded number of hidden units. The technical quality of the paper is of high standard, I did not observe any problems on this front. Except for - the point raised above about the (heuristic?) setting of the parameter alpha. - the functional derivative of L(q,theta) given in the appendix misses a term related to the L2 regularization. It re-appears in equation 9.

Confidence in this Review

2-Confident (read it all; understood it all reasonably well)


Reviewer 4

Summary

Interested in learning RBMs this paper has the interesting idea of writing the probability of the visible states in an RBM as an *expectation* over some distribution over hidden states. This then leads to a relaxation where one can learn this distribution over hidden states. A Frank-Wolfe algorithm can do this, which is quite attractive, though it still has the downside that MCMC must be used in each iteration.

Qualitative Assessment

Learning RBMs is important, but I think this paper has broader importance in that the ideas used here might be useful elsewhere as well. I think the immediate practical impact of this work is limited, but I still think this paper should be published because of the importance of the ideas. I have a few minor comments / questions. Firstly, in Proposition 3.1, the meaning of "both" should be clear-- does this mean jointly or separately. Secondly, the impact of fixing rather than optimizing alpha as done in the experiments could potentially affect convexity-- please discuss this. Thirdly, do you have any intuition for why the likelihoods are lower using only FW rather than CD in the experiments? (I have no idea why either should be higher, but this was somewhat surprising.) Fourth, in the experiments, a large number of parameters are set using validation data. Was this done using a grid search over all jointly or separately?

Confidence in this Review

2-Confident (read it all; understood it all reasonably well)


Reviewer 5

Summary

The paper considers a convex extension of RBM, allowing for RBMs with an infinite number of hidden units. The authors propose an algorithm based on the Frank-Wolfe algorithm for learning parameters which can be interpreted as iteratively adding a single hidden neuron to the current solution. The proposed model is evaluated in several experiments and the authors find that the taken approach can provide a good initialization for training RBMs (improving the performance over random initialization),

Qualitative Assessment

The paper is well written, easy to follow, and introduces an interesting extension to RBMs allowing for infinite hidden units. In particular, using the method for initialization of models for further training using CD seems promising. I do not have any critics regarding the theoretical part of the work. Regarding the experiments I have a few questions: * Do you have any idea why the models learned using your approach consistently perform worse than the models trained using CD? I guess this is due to the iterative process of (greedily) adding hidden units while not optimizing the "whole thing". Did you have a look at the filters identified in the first few iterations for MNIST? -- how do they look? What about the training likelihood? Are all models on the same level? * Figure 2 lacks error bars. Are the conclusions the same when looking at average performance? Minor comments: * L169: continues -> continue * L215: selected * L254: leading to ... / resulting in ...

Confidence in this Review

2-Confident (read it all; understood it all reasonably well)


Reviewer 6

Summary

This paper aims to address a very important problem, i.e., to determine the number of hidden units inside a RBM network. It gives an solution by introducing a prior on $\wv$ and solves the new optimization problem via Frank-Wolfe learning algorithm. The results are reasonable.

Qualitative Assessment

This paper is well written and aims to address an important problem. It gives a solution. However, following problems exist in this paper: 1) There is no convergence guarantee for the algorithm. 2) No criterion to stop adding new notes to $w$. The results can still only be obtained via cross-validation like method. For example, keep adding and testing the likelihood. This is different from the Chinese restaurant process, which can stop adding new tables. 3) The results are not strong enough to support the new framework. The difference from the proposed method and random initialization is not significant in Figure 2. === Post rebuttal The early stop is fine to me now. But it is not novel. I still suggest the authors to add stronger results to support the new framework.

Confidence in this Review

2-Confident (read it all; understood it all reasonably well)